# Nicotinamide Mononucleotide Supplementation Improves Mitochondrial Dysfunction and Rescues Cellular Senescence by NAD^+^/Sirt3 Pathway in Mesenchymal Stem Cells

**DOI:** 10.3390/ijms232314739

**Published:** 2022-11-25

**Authors:** Huan Wang, Yanan Sun, Chenchen Pi, Xiao Yu, Xingyu Gao, Chang Zhang, Hui Sun, Haiying Zhang, Yingai Shi, Xu He

**Affiliations:** The Key Laboratory of Pathobiology, Ministry of Education, College of Basic Medical Sciences, Jilin University, Changchun 130021, China

**Keywords:** MSC senescence, NMN, NAD^+^, Sirt3, mitochondrial function

## Abstract

In vitro expansion-mediated replicative senescence has severely limited the clinical applications of mesenchymal stem cells (MSCs). Accumulating studies manifested that nicotinamide adenine dinucleotide (NAD^+^) depletion is closely related to stem cell senescence and mitochondrial metabolism disorder. Promoting NAD^+^ level is considered as an effective way to delay aging. Previously, we have confirmed that nicotinamide mononucleotide (NMN), a precursor of NAD^+^, can alleviate NAD^+^ deficiency-induced MSC senescence. However, whether NMN can attenuate MSC senescence and its underlying mechanisms are still incompletely clear. The present study herein showed that late passage (LP) MSCs displayed lower NAD^+^ content, reduced Sirt3 expression and mitochondrial dysfunction. NMN supplementation leads to significant increase in intracellular NAD^+^ level, NAD^+^/ NADH ratio, Sirt3 expression, as well as ameliorated mitochondrial function and rescued senescent MSCs. Additionally, Sirt3 over-expression relieved mitochondrial dysfunction, and retrieved senescence-associated phenotypic features in LP MSCs. Conversely, inhibition of Sirt3 activity via a selective Sirt3 inhibitor 3-TYP in early passage (EP) MSCs resulted in aggravated cellular senescence and abnormal mitochondrial function. Furthermore, NMN administration also improves 3-TYP-induced disordered mitochondrial function and cellular senescence in EP MSCs. Collectively, NMN replenishment alleviates mitochondrial dysfunction and rescues MSC senescence through mediating NAD^+^/Sirt3 pathway, possibly providing a novel mechanism for MSC senescence and a promising strategy for anti-aging pharmaceuticals.

## 1. Introduction

Adult stem cells (ASCs), a promising source for regenerative medicine, have high proliferative potential and the capacity to differentiate into various cell types. Nevertheless, as individuals age, ASCs inevitably undergo cellular senescence [1,2], usually accompanied by mitochondrial dysfunction [3,4]. Mesenchymal stem cells (MSCs) are the early and in-depth study of ASCs, with a wide range of basic research and clinical application prospects [5]. Currently, MSCs possess great potentials for tissue healing and regenerative medicine because of the ability to self-renewal and multi-lineage differentiation. However, obtaining a sufficient number of MSCs needs long-term expansion in vitro, which inevitably results in MSC replicative senescence and then severely limits the applications in clinical therapy and regenerative medicine. Therefore, exploiting effective anti-aging pharmaceuticals and functional food to delay stem cell senescence is a crucial issue that we need to solve urgently.

Abnormal mitochondrial function could give rise to impaired mitochondrial integrity and biogenesis, the reduction in mitochondrial membrane potential (MMP) and adenosine triphosphate (ATP) production, and the enhanced reactive oxygen species (ROS) level, eventually accelerate cellular senescence or individual age [1,6,7,8,9]. Studies have shown that mitochondrial dysfunction is one of the hallmarks of aging [1,10]. Moreover, it has been confirmed that mitochondrial dysfunction and other cellular senescence-related pathways cooperate with each other and play an important role in the process of stem cell senescence [11,12]. Li et al. have recently reported that MSCs in aged mice had lower MMP, decreased nicotinamide adenine dinucleotide (NAD^+^)/ NADH ratio, and reduced ATP production as compared to young mice [13]. Consistent herewith, it has been verified that replicative senescence significantly increased the generation of ROS, decreased MMP and induced abnormal mitochondrial area and size in MSCs [14]. So far, the underpinning molecular mechanisms of mitochondrial dysfunction how to regulate stem cell senescence is still unclear.

NAD^+^, a key coenzyme in cellular energy metabolism, could be adaptive response to oxidative stress, involved in metabolic pathways and affect mitochondrial function [15]. NAD^+^ level determines the speed and degree of cellular senescence, and NAD^+^ depletion is closely related to energy metabolism disorder. In our previous study, we have demonstrated that NAD^+^ content significantly lowered in replicative senescent and natural senescent MSCs [16,17]. Consistently, elevated NAD^+^ synthesis is beneficial to delay MSC senescence [16]. Nicotinamide mononuclotide (NMN), the precursor of NAD^+^, can alleviate NAD^+^ deficiency-induced MSC senescence. It has been reported that NMN can increase the level of mitochondrial NAD^+^, normalize NAD^+^/NADH ratio, maintain mitochondrial homeostasis and then restore the function of muscle stem cells in aged mice [18]. Therefore, NMN has extensive guiding significance for the exploration of postponing stem cell senescence. 

Mammalian Sirtuins are homologs of Saccharomyces cerevisiae Sir2 (silent information regulator 2), and require NAD^+^ as a co-factor that drives deacetylase activity [19,20]. The accumulated evidence suggested that Sir2 or Sirtuins played crucial roles in regulating lifespan of organisms from unicellular yeast to mammals [21,22,23,24]. Among them, Sirt3 is the most thoroughly studied mitochondrial sirtuin, which can regulate various metabolic processes, including subunits of the electron transport chain (ETC), fatty acid oxidation, amino acid metabolism, redox balance, and the tricarboxylic acid (TCA) cycle [25,26]. Additionally, it has been reported that Sirt3-knockout mice had a shorter lifespan compared with wild-type mice [23]. Simultaneously, previous studies have shown that Sirt3 inhibited MSC natural senescence and oxidative stress-induced premature senescence through upregulating superoxide dismutase 2 (SOD2) expression and activity [27]. Moreover, Sirt3 also was implicated in a variety of age-related pathologies, such as cardiovascular diseases, osteoporosis, osteoarthritis, metabolic syndrome and neurodegenerative diseases [28,29]. Given our current understanding of Sirt3, its potential role in alleviating stem cell senescence by regulating mitochondrial function has yet to be unraveled.

As above, we have confirmed that NMN can attenuate NAD^+^ deficiency-induced MSC senescence. NAD^+^ repletion could reprogram dysfunctional stem cells and extend life span in mammals by improving mitochondrial function [30]. These findings suggested that NMN likely plays a pivotal role in the regulation of MSC senescence. In addition, as the NAD^+^ dependent deacetylase, Sirt3 has great influence on regulating mitochondrial function and stem cell senescence. Restoring mitochondrial NAD^+^ levels and Sirt3 activity enhanced reprogramming efficiency of aged somatic cells and extended the lifespan of human MSCs by delaying replicative senescence [31]. Thus, in the present study, we investigated that the regulatory effects and possible mechanisms of NMN on mitochondrial function and MSC senescence.

## 2. Results

### 2.1. Mitochondrial Dysfunction in Replicative Senescent MSCs

In the present study, MSCs in different passages were obtained by whole bone marrow adherent method and serial expansion in vitro. MSCs at early passage (EP) were long and fusiform in shape, exhibited good growth and distinct cell border, whereas late passage (LP) MSCs displayed senescence-like morphology such as irregular shapes, enlarged and flattened cell bodies (Figure 1a). The analytic results showed that the cell surface area was increased, whereas the cell aspect ratio significantly decreased with serial subcultivation (Figure 1a). In our previous study, cell surface antigens were used to identify obtained MSCs by flow cytometry. We found that serial expanded MSCs are positive for mesenchymal progenitor markers including CD44, CD90, and CD105, and negative for CD31 and CD45 [32]. Nevertheless, MSCs at LP presented attenuated osteogenesis compared with EP MSCs [17]. To further verify MSC senescence, SA-β-gal activity and senescence marker were detected. As expectedly, the number of blue-stained MSCs were elevated in LP MSCs and the analytic results indicated that SA-β-gal activity in LP MSCs was also significantly increased compared to that in EP MSCs (Figure 1b). Simultaneously, the senescence-associated biomarker P16^INK4a^ expression at mRNA level was up-regulated in LP MSCs (Figure 1c). Thus, LP MSCs displayed the senescent characteristics, and replicative senescence occurred with increasing passages. 

To identify the role of mitochondria in senescent MSCs, we firstly observed mitochondrial morphology from EP and LP MSCs by Mito-Tracker Red (MTR) fluorescence staining. As expectedly, the mitochondria in EP MSCs were in concentrated distribution, while those in LP MSCs were scattered and fragmented (Figure 1d). Moreover, transmission electron microscope revealed that the mitochondria in EP MSCs are elliptical, with complete double-layer membrane structure and continuous internal cristae; while the size of mitochondria in LP MSCs strikingly increased, with large vacuoles, missing double membrane structure and disrupted internal cristae (Figure 1e). The aforementioned results indicated that the mitochondrial morphology was abnormal along with in vitro expansion of MSCs. 

One of the hallmarks of aging is the occurrence of mitochondrial dysfunction [30]. To further understand the underlying molecular mechanisms of mitochondrial regulation on stem cell senescence, we explore the alterations of mitochondrial function. As shown in Figure 1f, the ATP content was significantly lower in LP MSCs compared to that in EP MSCs, while the intensity of ROS fluorescence was much stronger (Figure 1g). And the ROS levels were all decreased in EP and LP MSCs after treatment with N-Acetyl-cysteine (NAC, a ROS scavenger) (Appendix A). Then the LP MSCs mitochondrial dysfunction was further confirmed by the loss of MMP. As shown in Figure 1h, the green fluorescence was stronger, while the red fluorescence was weaker in LP MSCs than that in EP MSCs, and further quantitative analyses obtained similar results. The mitochondrial OCR in LP MSCs dominantly declined by the Seahorse XF96, as well as decreased basal respiration, maximum respiration and ATP production (Figure 1i). These observations supported the notion that mitochondrial dysfunction may be one of the destructive factors provoking MSC senescence.

### 2.2. Down-Regulation of NAD^+^/Sirt3 Signaling Pathway in LP MSCs

As a NAD^+^ dependent deacetylase, mitochondrial Sirt3 modulates the availability of NAD^+^ as a substrate to mitochondrial enzymes and may at least, in part attenuate cellular senescence through improving mitochondrial dysfunction [33]. To explore the possible mechanisms of ameliorating mitochondrial dysfunction and MSC senescence, the intracellular NAD^+^ content, NAD^+^/NADH ratio and Sirt3 expression were then examined. The results presented that NAD^+^ content (Figure 2a) and NAD^+^/NADH ratio (Figure 2b) were remarkably decreased in the LP MSCs in comparison to EP MSCs. The expression of Sirt3 was measured by RT-qPCR and Western blot analysis, respectively. Our data revealed that Sirt3 expression at both mRNA (Figure 2c) and protein levels (Figure 2d) were significantly reduced in LP MSCs. Overall, these data suggests that NAD^+^/Sirt3 signaling pathway was down-regulated, which may play a regulatory role in MSC senescence.

### 2.3. NMN Improves Mitochondrial Function and Inhibits Expansion-Mediated MSC Replicative Senescence 

To determine whether NMN, the precursor of NAD^+^, could ameliorate MSC senescence via improving mitochondrial function, we firstly added 100 μM NMN to LP MSCs for 24 h and analyzed mitochondrial morphology. The MTR staining showed that a more diffuse and less intense staining of the mitochondria in LP MSCs, while the distribution of mitochondria became concentrated after NMN treatment (Figure 3a). Then the improved mitochondrial function after NMN treatment was confirmed by increased ATP content (Figure 3b), decreased ROS level (Figure 3c) and up-regulated MMP (Figure 3d). Thus, NMN treatment could effectively normalize mitochondrial morphology and improve mitochondrial function in senescent MSCs.

To investigate whether NMN could influence MSC senescence, we examined the morphological characteristics and senescence-associated biomarker in LP MSCs after NMN treatment. The resulted displayed that NMN supplementation could alter the senescence-associated phenotypes caused by expansion in vitro. The cell body became more elongated and slenderer from flat, and the cell boundary gradually became clearer (Figure 3e). Meanwhile, the cell area prominently decreased, whereas the cell aspect ratio markedly increased after NMN addition (Figure 3e). Furthermore, SA-β-gal staining and the analytic results manifested the ratio of SA-β-gal-positive cells in LP MSCs after NMN treatment was markedly reduced (Figure 3f), and P16^INK4a^ mRNA expression was also significantly lower (Figure 3g). These results demonstrated that NMN exerts a protective effect on MSC senescence by ameliorating mitochondrial function.

### 2.4. NMN Up-Regulates NAD^+^/Sirt3 Signaling Pathway in Replicative Senescent MSCs

As a precursor of NAD^+^ synthesis, NMN supplementation could significantly increase the intracellular NAD^+^ content (Figure 4a) and the ratio of NAD^+^/NADH (Figure 4b) in LP MSCs. To clarify whether the up-regulation of NAD+ content caused by NMN repletion could affect Sirt3 expression, RT-qPCR and western blotting were performed accordingly. We found that Sirt3 expression at both mRNA (Figure 4c) and protein (Figure 4d) levels in LP MSCs treated with NMN presented a remarkable increase, suggesting that NMN replenishment might activate NAD^+^/Sirt3 signaling pathway in senescent MSCs.

### 2.5. Sirt3 Over-Expression Ameliorates Mitochondrial Function and Rescues MSC Senescence

In the light of lower Sirt3 expression in LP MSCs, we examined whether MSC senescence could be rescued by enforcing Sirt3 expression. For this purpose, LP MSCs were transduced with lentivirus-expressing Sirt3 (LV-Sirt3) and lentiviral vector (LV-Vector), followed by evaluating transduction efficiency through RT-qPCR and western blotting. Notably, enhanced green fluorescent proteins were successfully expressed in LV-Sirt3 and the LV-Vector group (Figure 5a). And Sirt3 expression at both the protein (Figure 5b) and mRNA (Figure 5c) levels was significantly up-regulated following Sirt3-expressing lentiviral transduction. Then MTR fluorescent staining was performed to assess mitochondrial morphology. As shown in Figure 5d, the more concentrated distribution and stronger staining of mitochondria were observed in LV-Sirt3 group in comparison with LV-Vector group. Moreover, the intracellular ATP content was significantly increased (Figure 5e), while the intensity of ROS fluorescence was much weaker in LV-Sirt3 group than that in LV-Vector group (Figure 5f), implying that Sirt3 over-expression in LP MSCs increased ATP content and reduced ROS production or accumulation. Furthermore, the mitochondrial OCR measured by the Seahorse XF96 was dramatically higher after increasing Sirt3 expression (Figure 5g). In consistent with these results, the basal respiration, maximal respiration capacity and ATP production of mitochondria were all elevated (Figure 5g). Collectively, these data suggested that Sirt3 over-expression in LP MSCs had a favorable regulatory effect on alleviating mitochondrial dysfunction.

In our previous study, we demonstrated that Sirt3 could attenuate MSC natural senescence and oxidative stress-induced premature senescence [27]. To further investigate the effects of Sirt3 in MSC replicative senescence, SA-β-gal activity and the expression of senescence-associated biomarker were investigated. As shown in Figure 5h, the rate of blue stained positive cells was seriously decreased in LV-Sirt3 group. What is more, P16^INK4a^ mRNA expression in LV-Sirt3 group were substantially inhibited compared to those in cells transduced with the vector (Figure 5i). These findings revealed that the mechanisms underlying Sirt3-regulated MSC replicative senescence may be linked to mitochondrial function.

### 2.6. Sirt3 Selective Inhibitor 3-TYP Induces Mitochondrial Dysfunction and Accelerates MSC Replicative Senescence 

Given that Sirt3 was highly expressed in EP MSCs, we wondered whether Sirt3 suppression has the adverse effects on mitochondrial function and MSC senescence. Thence, EP MSCs were treated with 100 μM 3-TYP or Vehicle for 24 h, and then the mitochondrial morphology was observed by MTR fluorescence staining. As shown in Figure 6a, the mitochondria are concentrated in EP MSCs, while the mitochondria were scattered and fragmented in EP MSCs after 3-TYP treatment, which indicated that 3-TYP could make the mitochondria morphology tend to be abnormal. In addition, mitochondrial dysfunction resulted from 3-TYP was also convinced by decreased intracellular ATP content (Figure 6b) and increased ROS level (Figure 6c). Furthermore, the loss of MMP was confirmed by JC-1 staining. As shown in Figure 6d, the green fluorescence was stronger while the red fluorescence was weaker in EP MSCs after 3-TYP treatment, implying Sirt3 selective inhibitor 3-TYP aggravated MMP doom. 

To clarify the effect of selectively inhibiting Sirt3 on the senescence of MSCs, classical senescence staining was performed. Representative cell images showed that EP MSCs after 3-TYP treatment exhibited enhanced expression of SA-β-gal (Figure 6e). The analytic results indicated that the ratio of SA-β-gal positive cells in EP MSCs after 3-TYP treatment was significantly elevated (Figure 6e). Moreover, the mRNA expression levels of the senescence markers P16^INK4a^ was greatly up-regulated in EP MSCs after 3-TYP treatment relative to young MSCs (Figure 6f). Taken together, our data showed that 3-TYP accelerating MSC replicative senescence may be closely associated with mitochondrial dysfunction mediated by Sirt3 inhibition.

### 2.7. NMN Rejuvenates Mitochondrial Function and Delays MSC Senescence through Activating NAD^+^/Sirt3 Signaling Pathway

To gain deeper insights into the mechanism underlying how NMN rejuvenated mitochondrial function and delayed MSC senescence, we next detected mitochondrial function following NMN and 3-TYP co-treatment in LP MSCs. As shown in Figure 7a, in comparison with LP MSCs, NMN supplementation in LP MSCs resulted in more normal mitochondrial morphology and more concentrated distribution, whereas mitochondria were scattered and fragmented in response to 3-TYP treatment. What is more, 3-TYP could not only significantly down-regulate the increased intracellular ATP content (Figure 7b), but also reverse the decreased ROS levels caused by NMN in LP MSCs (Figure 7c). Simultaneously, NMN repletion increased the MMP in LP MSCs, while 3-TYP treatment could weaken the increased MMP caused by NMN (Figure 7d). To further prove that MSC replicative senescence is relevant to NAD^+^-Sirt3 signaling, we assessed the effects of NMN and 3-TYP co-treatment on senescent-associated phenotypes. SA-β-gal staining and analytic results revealed that the percentages of SA-β-gal-positive cells was strikingly decreased in LP after NMN repletion, while 3-TYP treatment could resume abundant SA-β-gal activity (Figure 7e). In addition, 3-TYP significantly up-regulated NMN-induced low P16^INK4a^ expression at the mRNA level (Figure 7f). The above results indicated the precursor of NAD^+^ synthesis-NMN might inhibit MSC senescence by improving mitochondrial function through NAD^+^/Sirt3 pathway.

Next, we also assessed mitochondrial function and MSC senescent-associated phenotypes in EP MSCs after 3-TYP and NMN co-treatment. Of note, mitochondria distribution became dispersed and fragmented in EP MSCs after 3-TYP treatment, while mitochondria were concentrated in EP MSCs co-treated with 3-TYP and NMN, which is similar to those in EP MSCs (Figure 8a). In addition, 3-TYP treatment reduced intracellular ATP content (Figure 8b), increased the ROS levels (Figure 8c) and decreased the MMP (Figure 8d) in EP MSCs, whereas NMN repletion restored higher ATP content (Figure 8b), lower ROS levels (Figure 8c) and more distinct MMP (Figure 8d). Furthermore, in comparison with EP MSCs, the ratios of SA-β-gal-positive cells (Figure 8e) and the mRNA expression of senescence-associated biomarkers P16^INK4a^ (Figure 8f) displayed obviously upward trend after 3-TYP treatment. On the contrary, NMN repletion could restore MSC rejuvenation (Figure 8e,f). Taken together, these findings revealed that the mechanisms underlying NMN-regulated mitochondrial function and MSC replicative senescence were concerned with the NAD^+^/Sirt3 signaling pathway. 

## 3. Discussion

MSCs are the early and in-depth study of adult stem cells, with a wide range of basic research and clinical application prospects. Importantly, MSCs are the safe and feasible source for tissue healing and regenerative medicine at present. Unfortunately, the function of MSCs is known to decline with expansion in vitro. This process may be implicated in the loss of maintenance of tissue homeostasis, further resulting in age-related organ failure and even diseases. Our previous studies indicated that senescent MSCs have evident morphological alterations, declined cell proliferation, attenuated pluripotency and irreparable cell cycle arrest [27,34]. Consistent herewith, our results showed that LP MSCs displayed senescence-like morphology, including irregular shapes, enlarged and flattened cell bodies. At the same time, SA-β-gal activity and the expression of senescence marker P16^INK4A^ also displayed significant up-regulated trends in LP MSCs compared to EP MSCs. Therefore, exploiting effective anti-aging drugs and functional food to delay stem cell senescence deserves further investigation.

Mitochondria, the main site of oxidative phosphorylation (OXPHOS), not only could synthesize ATP to provide direct energy for vital activities, but also play an irreplaceable role in metabolism, cellular homeostasis, and antioxidant function [35]. Mitochondrial function is linked to stem cell maintenance and activation [30]. Ample evidence suggested that mitochondrial dysfunction was one of the characteristics of aging [1,10]. Ordinarily, mitochondrial dysfunction is characterized by impaired mitochondrial integrity and biogenesis, increased ROS level, decreased ATP synthesis, low MMP and reduced OCR level [1,6,7,8,9]. Consistent herewith, Lee et al. verified that replicative senescence seriously increased the generation of ROS, decreased MMP and induced abnormal mitochondrial area and size in MSCs [14]. It was also reported that mitochondrial dysfunction in muscle stem cells from aged mice was probed by the loss of MMP, a reduction in cellular ATP concentrations, decreased oxidative respiration rates and down-regulated expression of TCA cycle gene [30]. In accordance to these data, herein, we unveiled that the mitochondrial morphology and structure were obviously injured in senescent LP MSCs compared to that in young EP MSCs. Simultaneously, mitochondrial function such as the ATP content, MMP and the OCR level were dominant declined, while the levels of ROS increased in senescent LP MSCs. Thus, we speculated that mitochondrial dysfunction may act as a driver of MSC senescence. However, the methods for detecting mitochondrial function are relatively simple and not up-to-date, which is the shortcoming of this study. In subsequent work, methods such as flow cytometry and tetramethylrhodamine ethyl ester perchlorate (TMRE) or tetramethylrhodamine methy ester (TMRM) will be considered.

The mammalian “NAD^+^ world” senescence theory holds that the level of NAD^+^ determines the speed and degree of senescence. Besides, NAD^+^ could be an adaptive response to oxidative stress, involved in metabolic pathways and affect mitochondrial function [15]. In our previous study, we have demonstrated that NAD^+^ content significantly lowered in replicative and natural senescent MSCs, and elevated NAD^+^ synthesis is beneficial to delay MSC senescence [16,17]. Consistently, the current results also found NAD^+^ content and NAD^+^/NADH ratio were decreased as MSC expansion in vitro. As the precursor of NAD^+^, NMN can increase the level of mitochondrial NAD^+^, normalize NAD^+^/NADH ratio, maintain mitochondrial homeostasis and then restore the function of muscle stem cells in aged mice [18]. In addition, NMN has an effective influence on improving mitochondrial dysfunction, reducing ROS accumulation, repairing DNA damage, and compensating for defects in cell survival caused by NAD^+^ deficiency [36,37]. In accordance with these data, herein, we unveiled that NMN repletion in LP MSCs up-regulated the decreased NAD^+^ content and NAD^+^/NADH ratio and also ameliorated damaged mitochondrial morphology and function, including increased ATP content, reduced ROS level, higher MMP and enhanced OCR level. The above descriptions confirm that mitochondrial dysfunction is one of the characteristics of MSC senescence. At the same time, exploring the pharmaceuticals or nutritional diets acting on mitochondria has non-negligible significance in the treatment of age-related diseases and extension of life-span.

NMN, the precursor of NAD^+^, can alleviate NAD^+^ deficiency-induced MSC senescence. Our published results indicated exogenous supplement of NMN could delay MSC senescence by NAD^+^/Sirt1 pathway [16]. NAD^+^ precursor molecules, such as nicotinamide (NAM) or nicotinamide riboside (NR), could effectively prevent or reverse the age-related NAD^+^ decline [38]. Moreover, NAD^+^ levels and energy metabolism can be improved by NMN administration in high-fat diet-induced diabetes and elderly mouse models [39]. Peter Belenky et al. have recently reported that NMN inhibited MSC senescence in aged mice by restoring mitochondrial homeostasis [40]. In congruent with these data, NMN repletion in LP MSCs could make the cell morphology tend to be young, decrease the cell surface area and increase the aspect ratio. Simultaneously, SA-β-gal activity and the expression of senescence marker P16^INK4A^ were notably shrunk in LP MSCs after MNM treatment than those in LP MSCs. Additionally, NMN repletion effectively promoted MSC expansion in vitro and in vivo [41]. Importantly, the expanded MSCs had heightened osteogenesis, but reduced adipogenesis [41]. What is more, calorie and dietary restrictions, administration of NMN and/or NAD^+^ might extend life-span and health-span [42]. These observations support the notion that NMN safeguards the morphological and functional integrity of mitochondria during MSC expansion in vitro.

As the NAD^+^ dependent deacetylase, Sirt3 is the most thoroughly studied mitochondrial sirtuin and regulates various metabolic processes. NAD^+^ depletion can decrease the expression of Sirt3 and increase the acetylation level of its target proteins. Since NMN supplementation can facilitate NAD^+^ content and the NAD^+^/NADH ratio, whether NMN might regulate the expression of Sirt3 by promoting NAD^+^ synthesis. On account of these, we detected the expression of Sirt3 in MSC replicative senescence. The results displayed that Sirt3 expression at mRNA and protein level were markedly lowered in LP MSCs, while NMN supplementation reversed the reduction in Sirt3 expression caused by replicative senescence. In a study performed by Denu, Sirt3 over-expression in LP MSCs reduced oxidative stress, enhanced their ability to differentiate, and furthermore ameliorated age-related senescence [43]. Besides, it has been reported that Sirt5 and Sirt3 protect cells from oxidative stress, while Sirt4 exacerbates oxidative stress [44]. Their roles and interactions within mitochondria are quite complicated. So further research will be critical for exploring the compensation function among mitochondrial sirtuins and continuing to approach actual clinical practice. Targeting Sirt3 to improve mitophagy may protect against MSC senescence and senile osteoporosis [45]. Consistent with these data, a preliminary work has found that over-expression of Sirt3 could reduce ROS level, alleviate DNA injury and partly reverse the senescence-associated phenotypic features in MSC natural and oxidative stress-mediated premature senescence. Similarly, we herein manifested that Sirt3 over-expression rescued expansion-mediated MSC replicative senescence. Furthermore, we tested the mitochondrial function and MSCs senescence in EP MSCs with Sirt3 selective inhibitor 3-TYP. Inconsistent with Liu’s research [46], we chose 100 μm as the concentration of 3-TYP in the study, which may be related to the short duration of action and the characteristics of young and senescent MSCs. The results showed that 3-TYP evidently enhanced SA-β-gal activity and the expression of senescence marker P16^INK4A^ in young EP MSC. These results indicate that absence of Sirt3 results in MSC senescence-associated molecular functional alterations. Mitochondrial function and cellular senescence will be investigated in Sirt3 knockdown or CRISPR/Cas9-mediated Sirt3 knockout MSCs in the future study.

Moreover, many studies have further discovered that Sirt3 had essential regulation on mitochondrial function, including mitochondrial electron transport, the activity of mitochondrial oxidative respiratory chain complex I and mitochondrial ATP production [47,48,49,50]. It has been demonstrated that NMN supplementation can rescue the accumulation of NADH, inactivation of mitochondrial Sirt3, and loss of mitochondrial electron transport chain complex I incited by mitochondrial NAD^+^ depletion [51,52]. In congruent with these statements, Sirt3 over-expression in LP MSCs markedly improved mitochondrial function confirmed by more normal mitochondrial morphology, elevated ATP content and OCR, as well as reduced ROS levels. On the contrary, 3-TYP administration in EP MSCs greatly impaired mitochondrial function. To further verify that NMN supplementation could promote mitochondrial function and alleviate senescence through NAD^+^/Sirt3 signal pathway, functional recovery experiments were performed. The results indicated that NMN supplementation alleviated mitochondrial morphological and functional abnormalities induced by Sirt3 inhibition and partially reversed MSC senescence. Son et al. discovered restoring mitochondrial NAD^+^ levels and Sirt3 activity by over-expressing nicotinamide nucleotide transhydrogenase (NNT) and nicotinamide mononucleotide adenylyltransferase 3 (NMNAT3) and enhanced reprogramming efficiency of aged somatic cells, and then extended the life-span of human MSCs by delaying replicative senescence [31]. In addition, CD38 is required for improving the age-related NAD decline and mitochondrial dysfunction by modulation of Sirt3 activity [33]. To sum up, NMN repletion ameliorated mitochondrial dysfunction and alleviate MSC senescence via the NAD^+^ /Sirt3 pathway. 

## 4. Materials and Methods

### 4.1. MSC Isolation and Subculture

MSCs were obtained according to a previously described method [16,17,32]. Primary MSCs derived from healthy, 1–2-month-old male Wistar rats were isolated by the whole bone marrow adherent method. Briefly, cell culture medium contained 89% Dulbecco’s Modified Eagle’s medium with nutrient mixture F-12 (DMEM-F12; Gibco, New York, NY, USA), 10% heat-inactivated fetal bovine serum (FBS, Gibco, New York, NY, USA), and 1% penicillin/ streptomycin solution (Gibco, New York, NY, USA), and then replaced every 2–3 days. After reaching 80% confluence, cells were detached by 0.25% trypsin-EDTA (Gibco, New York, NY, USA) and expanded at a ratio of 1:3. MSCs at early passage 3, termed as the EP (early passage, EP) MSCs. And MSCs at late passages 10 (late passage, LP) used in the subsequent experiments were acquired from young P3MSCs via successive passages.

### 4.2. Quantitative Analysis of MSC Morphology

Firstly, MSC morphology was observed under a high magnification (×200) microscope, and then multiple fields were selected to collect images. Next, the cell surface area and the longest and the shortest diameters passing through the nucleus of a single MSC were calculated by CellEntry software. Then the ratio of the longest diameter to the shortest diameter was the cell aspect ratio. Finally, the obtained cell surface area and cell aspect ratio were analyzed. 

### 4.3. Senescence-Associated β-Galactosidase (SA-β-gal) Staining

Senescent cell histochemical staining kit (Beyotime, Shanghai, China) was applied to evaluate SA-β-gal activity. Cells were fixed in fixation buffer at RT for 15 min, washed twice with phosphate-buffered saline (PBS), added to the pre-prepared staining solution mixture, and incubated in a CO_2_-free incubator from light at 37 °C for 12–24 h. The reaction was stopped by PBS. Then, the percentage of β-galactosidase-positive cells was determined using a bright-field microscope (OLYMPUS, Tokyo, Japan) for statistical analysis.

### 4.4. Gene Expression Analysis

Total RNA extracted from MSCs using Trizol (Takara, Beijing, China) was reverse transcribed to the complementary DNA (cDNA) with the Reverse transcription kit. Real-Time Quantitative PCR (RT-qPCR) was performed by the ABI 7300 Real-Time PCR System with SYBR green incorporation. Rat-specific primers were synthesized and listed as follows: β-actin: forward 5′-GGAGATTACTGCCCTGGCTCCTA3′, reverse 5′-GACTCATCGTACTCCTGCTTGCTG-3′; Sirt3:forward 5′-GGTAGAAAAAGCAACCACGAAGC-3′, reverse 5′-ACATAAACCTCTGTCT-GTGATGCC-3′; P16^INK4a^: forward 5′-AAACACTTTCGGTCGTACCC-3′, reverse 5′-GTCCTCGCAGTTCGAATC-3′. The thermal cycling protocol included pre-incubation at 95 °C for 2 min, followed by 40 cycles of amplification at 95 °C for 15 s and annealing for 30 s at 60 °C, and a final extension at 72 °C for 5 min. β-actin was amplified as a reference gene to normalize the relative expression of mRNA using the 2^−ΔΔCt^ cycle threshold method.

### 4.5. Mitochondrial MitoTracker Red Staining

When the confluence reached 80–90%, cells were wash with PBS 3 times, and an appropriate amount of preheated (37 °C) MitoTracker^®^ Red (Gibco, New York, NY, USA) probe staining solution was added. Then incubated in a 37 °C, 5% CO_2_ incubator for 15–45 min. After staining, wash 2–3 times with PBS, replace the staining solution with culture medium or buffer, and then place it under a fluorescence microscope for observation or a fluorescence microplate reader for reading.

### 4.6. Measurement of ATP Levels

ATP levels were measured with the ATP Assay Kit (Beyotime, Shanghai, China). 5 × 10^3^ MSCs were seeded in each well of 6-well plates with complete medium. Dilute the ATP standard solution with ATP detection lysate to an appropriate concentration gradient, add 20 μL sample or standard diluted in concentration gradient to the sample hole or standard curve hole respectively, mix well, and measure with a chemiluminometer (luminometer) RLU value, calculate the content of ATP in the sample according to the standard curve.

### 4.7. Measurement of ROS Levels

About 3 × 10^5^ cells were harvested, washed with serum-free medium and incubated with 5 μmol/L dihydroethidium (DHE, Beyotime, Shanghai, China) at 37 °C for 30 min. Then the cells were harvested, washed and resuspended in serum-free culture medium. After staining, wash 2–3 times with PBS, replace the staining solution with fresh culture medium or buffer, and then place it under a fluorescence microscope (excitation 300 nm and emission 610 nm) (OLYMPUS, Tokyo, Japan) for observation and quantification with ImageJ software. Besides, young and senescent MSCs cultured with 10 mM N-Acetyl-cysteine (NAC, a ROS scavenger) (Selleck, Houston, TX, USA) for 24 h to test the specify of DHE [53]. 

### 4.8. Measurement of MMP

The MSCs were planted in 24-well plates at a density range of 1 × 10^4^–3 × 10^4^ /well, then these cells were used for measuring ∆Ψm. The ∆Ψm was measured by a JC-1 kit (Beyotime, Shanghai, China) following the manufacturer’s instructions. MSCs were rinsed with PBS and incubated with JC-1 staining solution at 37 °C for 20 min. Then, the inverted fluorescence microscope (Olympus, Tokyo, Japan) was used to capture the pictures and calculate the ratio of the red: green fluorescence.

### 4.9. Oxygen Consumption Rate (OCR)

MSCs were plated in a density of 2 × 10^4^ cells per well on Seahorse XF96 polystyrene tissue culture plates (Seahorse Bioscience, North Billerica, MA, USA). Then, Oligomycin (2 μM), Carbonyl cyanide 4-trifluoromethoxy-phenylhydrazone (FCCP) (0.25 μM), Antimycin A (0.5 μM) and Rotenone (3 μM) were delivered to detect the spare respiration capacity, maximal respiration and ATP productivity, respectively.

### 4.10. Determination of Intracellular NAD^+^ and NAD^+^/NADH Ratio

Intracellular NAD^+^ content and NAD^+^/NADH ratio were measured with the NAD^+^/NADH Quantification Colorimetric Kit (BioVision, San Francisco, CA, USA), as recommended by manufacturer’s instructions. Total 2 × 10^5^ cells were harvested and sorted directly into 200 μL of lysis buffer by NADH/NAD^+^ Extraction Buffer. Then, the optical density at OD 450 nm was read using a multi-well spectrophotometer. Final NAD^+^ and NADH content, and NAD^+^/NADH ratio in each sample were calculated in accordance with the standard curve created by NADH standards from the kit. And the data obtained were normalized to the total cell number.

### 4.11. Lentiviral Transduction of MSCs

Prior to lentiviral transduction, MSCs were seeded into 24-well plates at a density of 1.5 × 10^4^ per well. Then, the cells were transduced with the purchased lentiviral particles encoding Sirt3 and its control LV-Vector (GeneChem, Shanghai, China), when reached 40–50% confluence. The EGFP expression was monitored under a fluorescence microscope (OLYMPUS, Tokyo, Japan) after 72–96 h. And transduction efficiency was evaluated by RT-qPCR and Western blot.

### 4.12. NMN and 3-TYP Intermediates Treatment

Senescent LP MSCs were cultured in the presence or absence of 100 µM NMN (PubChem CID: 16219737) for 48 h. To determine the optimal concentration of 3-TYP (PubChem CID: 9833992) without inducing robust cellular toxicity, 3.0 × 10^3^ cells were seeded in 96-well plates and incubated in a humidified incubator at 37 °C for 24 h. Thereafter, cells were treated with different concentrations of 3-TYP (0–200 µM; MCE; Monmouth Junction, NJ, USA) for 72 h before the addition of CCK8 solution (10 µL/well). Young EP MSCs were treated with complete medium containing 100 µM 3-TYP or PBS for 12 h.

### 4.13. Western Blot Analysis

Total protein was extracted using RIPA lysis buffer (Beyotime, Shanghai, China) supplemented with proteinase inhibitors. Protein concentration was determined by a BCA Protein Assay Kit (Beyotime, Shanghai, China). Then, 25 μg of protein sample was separated by 12% SDS-PAGE and transferred onto PVDF membranes (Burlington, MA, USA) by electroblotting, after which nonspecific binding to the membrane was blocked with 5% non-fat milk for 1–2 h at RT. Then, the blotted membranes were probed with anti-Sirt3 (1:2000, Abcam, Cambridge, MA, UK) and anti-β-actin (1:1500, Abcam, Cambridge, MA, UK) antibodies diluted in Tris-buffered saline (TBS) overnight at 4 °C. After incubating with antirabbit IgG secondary antibody (1:2000 dilution, Proteintech, Chicago, IL, USA), protein blots were visualized using an Electro-Chemi-Luminescence detection system (JENE, UK) and quantified with ImageJ software.

### 4.14. Statistical Analysis

All experiments were performed three times independently and data were presented as mean ± standard deviation. Statistical significance between groups was determined by using a two-tailed Student’s *t* test or a one-way ANOVA. *p* values indicated thusly: * *p* < 0.05, ** *p* < 0.01, and *** *p* < 0.001 vs. EP were considered statistically significant. 

## 5. Conclusions

In conclusion, the present study systematically elucidated that replicative senescent MSCs exhibit mitochondrial dysfunction, including abnormal mitochondrial morphology and structure, decreased intracellular ATP content, increased ROS level, MMP loss, and reduced OCR. NMN repletion could rejuvenate mitochondrial function and attenuate MSC senescence by activating NAD^+^/Sirt3 signaling pathway (Figure 9). Our findings may not only enrich insights into the molecular mechanisms underlying stem cell senescence from the perspective of energy metabolism, but also potentially provide a promising preparation strategy for anti-aging pharmaceuticals and functional food. Nevertheless, the complex mechanisms of NMN regulation of MSC senescence and its role in vivo await further confirmation. Deeper studies on whether NMN can act as an anti-aging drugs or functional food will be the focus of our future studies.

## Figures and Tables

**Figure 1 ijms-23-14739-f001:**
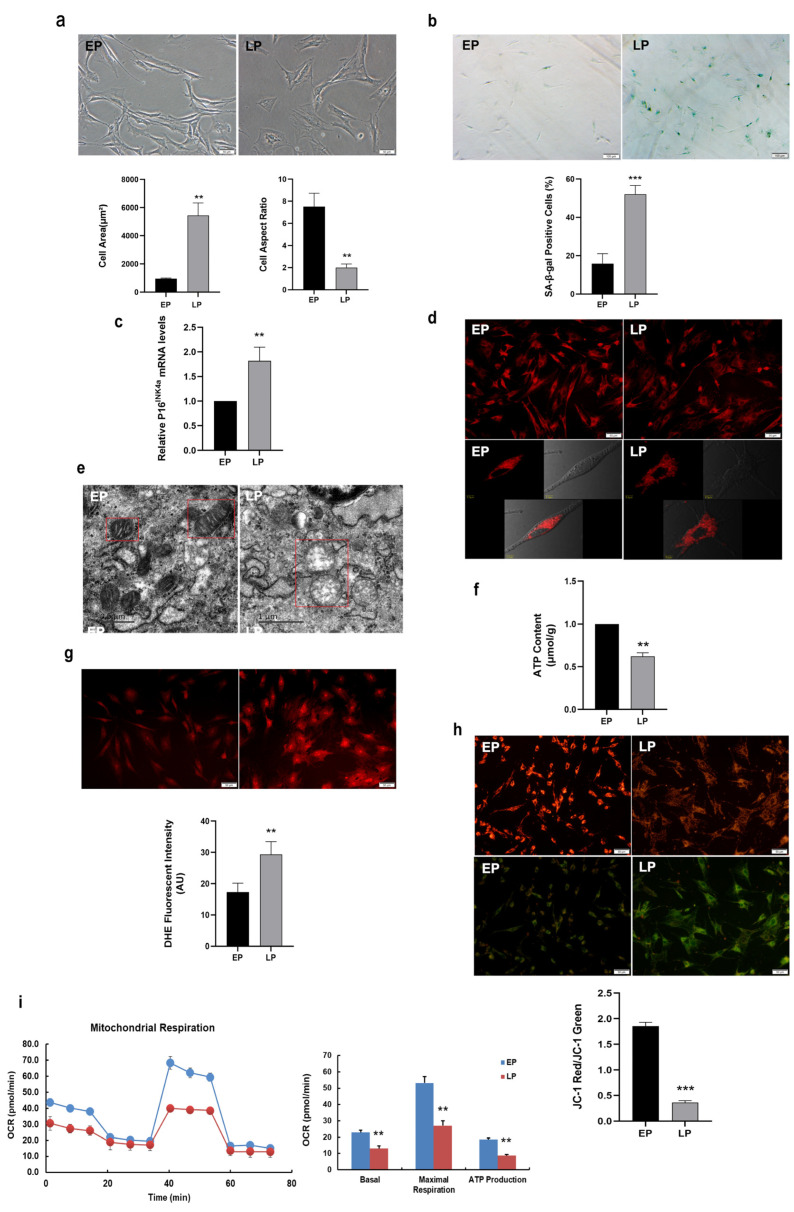
Senescence-associated variations and mitochondrial dysfunction in replicative senescent mesenchymal stem cells (MSCs). (**a**) Morphological characteristics of young MSCs at early passage (EP) and replicative senescent MSCs at late passage (LP) (scale bar = 50 µm), and analysis of cell surface area and cell aspect ratio; (**b**) Senescence-associated β-galactosidase (SA-β-gal) staining (scale bar = 100 µm) and quantitative analysis of SA-β-gal-positive cells. (**c**) The mRNA expression of senescence-associated biomarker P16^INK4a^. (**d**) Mito-tracker red (MTR) fluorescent staining to observe the mitochondrial morphology. (**e**) The ultrastructure of mitochondrion by transmission electron microscope (scale bar = 0.5 µm). (**f**) Adenosine triphosphate (ATP) content. (**g**) The reactive oxygen species (ROS) levels determined by dihydroethidium (DHE) staining (scale bar = 50 μm). (**h**) JC-1 staining (scale bar = 50 μm) to detect mitochondrial membrane potential (MMP). (**i**) Determination of the oxygen consumption rate (OCR) level. Data are expressed as mean ± SD, *n* = 3, ** *p* < 0.01, *** *p* < 0.001.

**Figure 2 ijms-23-14739-f002:**
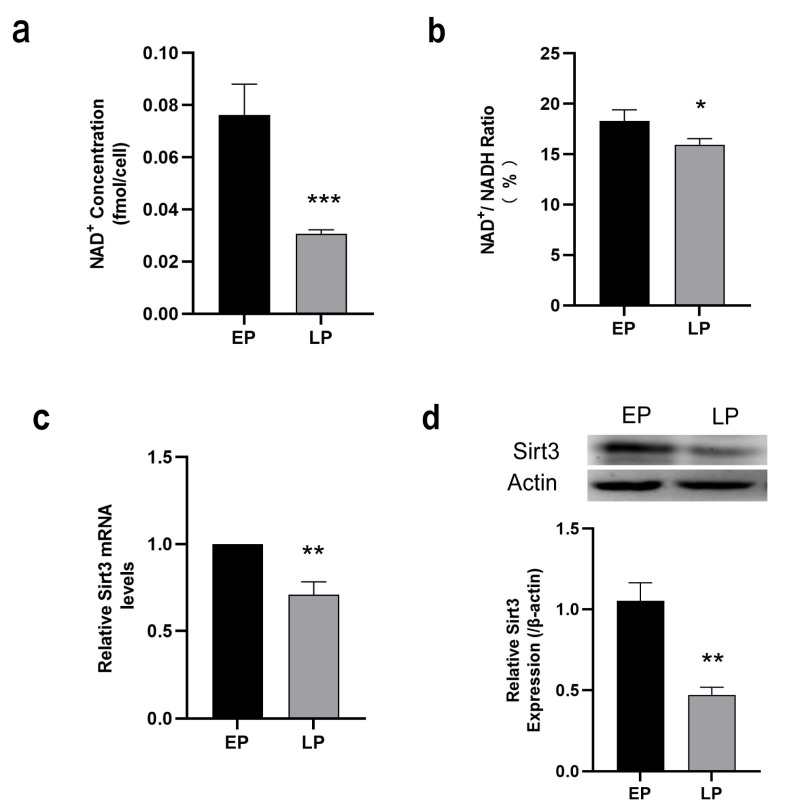
Measurement of nicotinamide adenine dinucleotide (NAD^+^)/Sirt3 signaling pathway in late passage mesenchymal stem cells (LP MSCs). (**a**) The intracellular NAD^+^ concentration. (**b**) NAD^+^/NADH ratio. (**c**) Examination of Sirt3 mRNA expression by RT-qPCR. (**d**) Sirt3 protein expression was detected by western blotting. Data are expressed as mean ± SD, *n* = 3, * *p* < 0.05, ** *p* < 0.01, *** *p* < 0.001.

**Figure 3 ijms-23-14739-f003:**
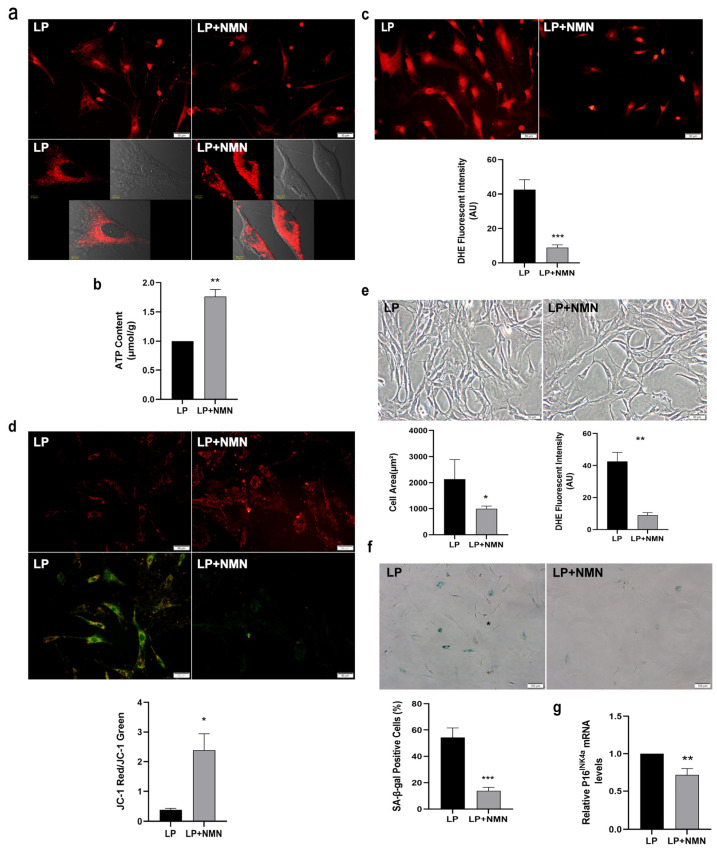
Nicotinamide mononucleotide (NMN) repletion improves mitochondrial function and prevents mesenchymal stem cell (MSC) replicative senescence. (**a**) Mito-tracker red (MTR) fluorescent staining to observe the mitochondrial morphology (scale bar = 50 μm). (**b**) Adenosine triphosphate (ATP) content. (**c**) Intracellular reactive oxygen species (ROS) levels were determined using dihydroethidium (DHE) staining (scale bar = 50 μm). (**d**) The detection of mitochondrial membrane potential (MMP) by JC-1 staining (scale bar = 50 μm). (**e**) Morphological characteristics of MSCs (scale bar = 50 μm) and analysis of cell surface area and cell aspect ratio. (**f**) Senescence-associated β-galactosidase (SA-β-gal) staining (scale bar = 100 µm) and quantitative analysis of SA-β-gal-positive cells. (**g**) Gene expression of senescence-associated biomarker P16^INK4a^. Data are expressed as mean ± SD, *n* = 3, * *p* < 0.05, ** *p* < 0.01, *** *p* < 0.001.

**Figure 4 ijms-23-14739-f004:**
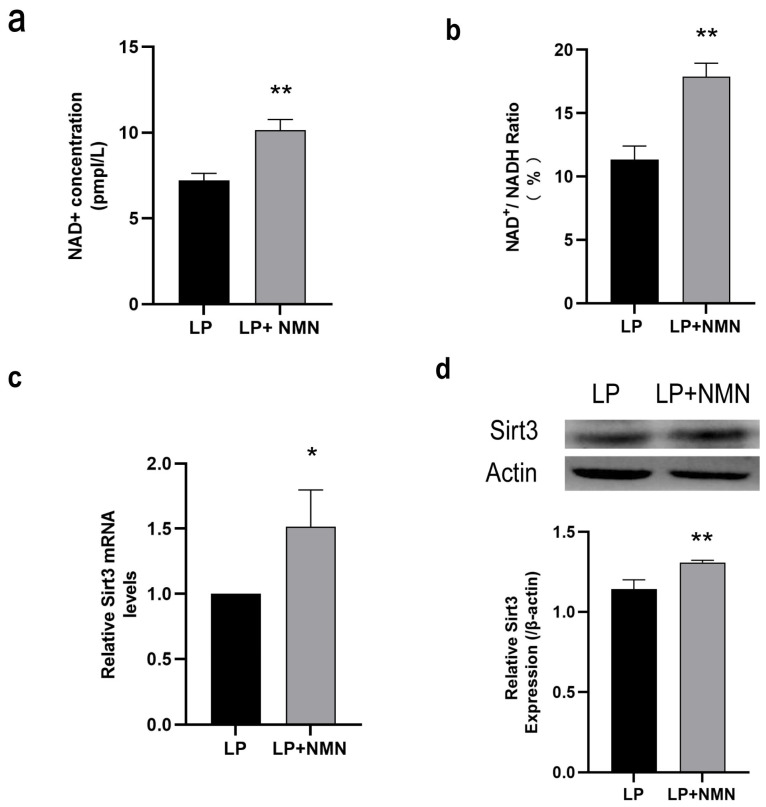
The nicotinamide adenine dinucleotide (NAD^+^)/Sirt3 signaling pathway is up-regulated in late passage mesenchymal stem cells (LP MSCs) treated by exogenous nicotinamide mononucleotide (NMN). (**a**) The intracellular NAD^+^ concentration. (**b**) NAD^+^/NADH ratio. (**c**) Sirt3 mRNA expression determined by RT-qPCR. (**d**) Sirt3 protein expression determined by western blot. Data are expressed as mean ± SD, *n*= 3, * *p* < 0.05, ** *p* < 0.01.

**Figure 5 ijms-23-14739-f005:**
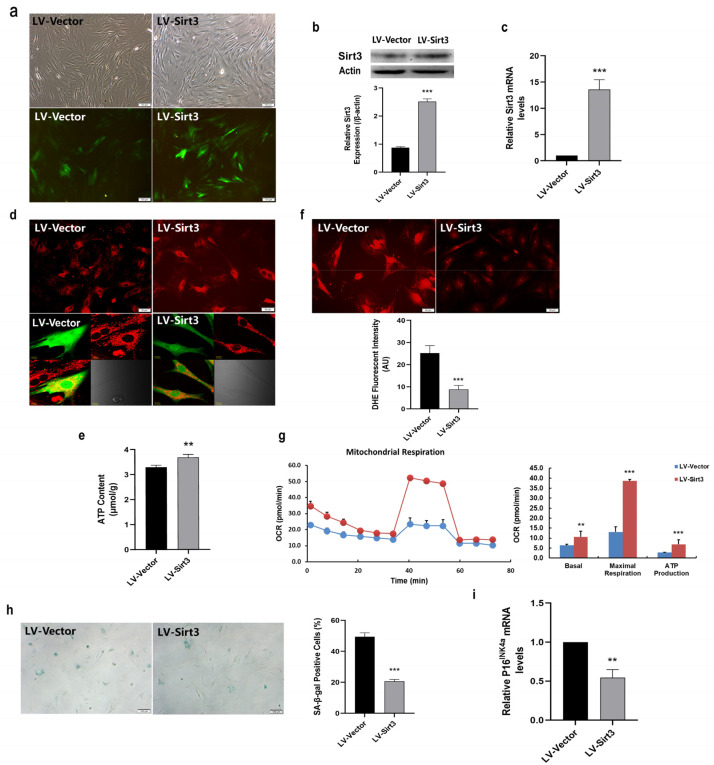
Sirt3 over-expression ameliorates mitochondrial function and rescues mesenchymal stem cell (MSC) senescence. (**a**) Fluorescence images showed that Sirt3 was successfully over-expressed in late passages (LP) MSCs (scale bar = 100 μm). (**b**) Sirt3 protein expression determined by western blot. (**c**) Examination of Sirt3 mRNA expression by RT-qPCR. (**d**) Mito-tracker red (MTR) fluorescent staining to observe the mitochondrial morphological features (scale bar = 50 μm). (**e**) Adenosine triphosphate (ATP) content. (**f**) Dihydroethidium (DHE) fluorescence intensity was detected by microphotography (scale bar = 50 μm). (**g**) Determination of the oxygen consumption rate (OCR) level. (**h**) Senescence-associated β-galactosidase (SA-β-gal) staining (scale bar = 100 µm) and quantitative analysis of β-gal-positive cells. (**i**) Examination of P16^INK4a^ mRNA expression by RT-qPCR. Data are expressed as mean ± SD, *n*= 3, ** *p* < 0.01, *** *p* < 0.001.

**Figure 6 ijms-23-14739-f006:**
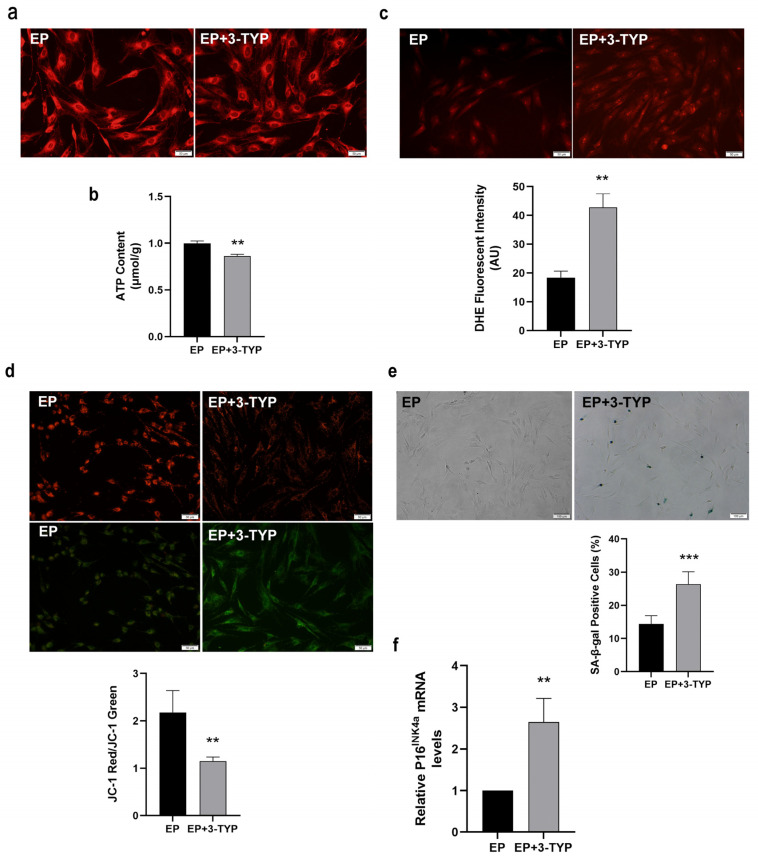
The effects of 3-TYP on mitochondrial function and mesenchymal stem cell (MSC) senescence. (**a**) Mitochondrial morphological features (scale bar = 50 μm). (**b**) Adenosine triphosphate (ATP) content. (**c**) Intracellular reactive oxygen species (ROS) levels were determined using dihydroethidium (DHE) staining (scale bar = 50 μm). (**d**) JC-1 staining (scale bar = 50 μm) were to detect mitochondrial membrane potential (MMP). (**e**) Senescence-associated β-galactosidase (SA-β-gal) staining (scale bar = 100 μm) and quantitative analysis of β-gal positive cell rate. (**f**) Senescence-associated biomarker P16^INK4a^ mRNA expression by RT-qPCR. Data are expressed as mean ± SD, *n*= 3, ** *p* < 0.01, *** *p* < 0.001.

**Figure 7 ijms-23-14739-f007:**
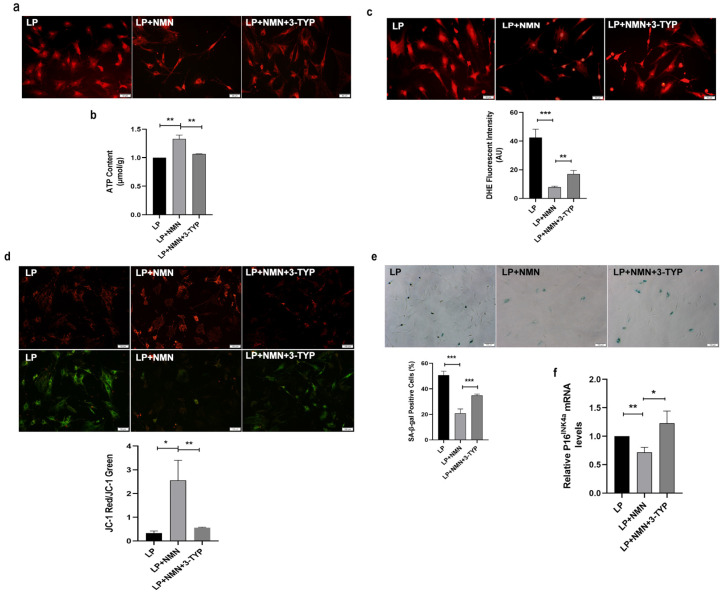
3-TYP aggravates nicotinamide mononucleotide (NMN)-improved mitochondrial dysfunction and cellular senescence in late passages mesenchymal stem cells (LP MSCs). (**a**) Observation of mitochondrial morphology by mito-tracker red (MTR) fluorescent staining (scale bar = 50 μm). (**b**) Adenosine triphosphate (ATP) content. (**c**) The reactive oxygen species (ROS) levels tested by dihydroethidium (DHE) staining (scale bar = 50 μm) and quantitative analysis. (**d**) The detection of mitochondrial membrane potential (MMP) (scale bar = 50 μm). (**e**) Senescence-associated β-galactosidase (SA-β-gal) staining (scale bar = 100 μm) and quantitative analysis of the positive cell rate. (**f**) Examination of P16^INK4a^ mRNA expression by RT-qPCR. Data are expressed as mean ± SD, *n* = 3, * *p* < 0.05, ** *p* < 0.01, *** *p* < 0.001.

**Figure 8 ijms-23-14739-f008:**
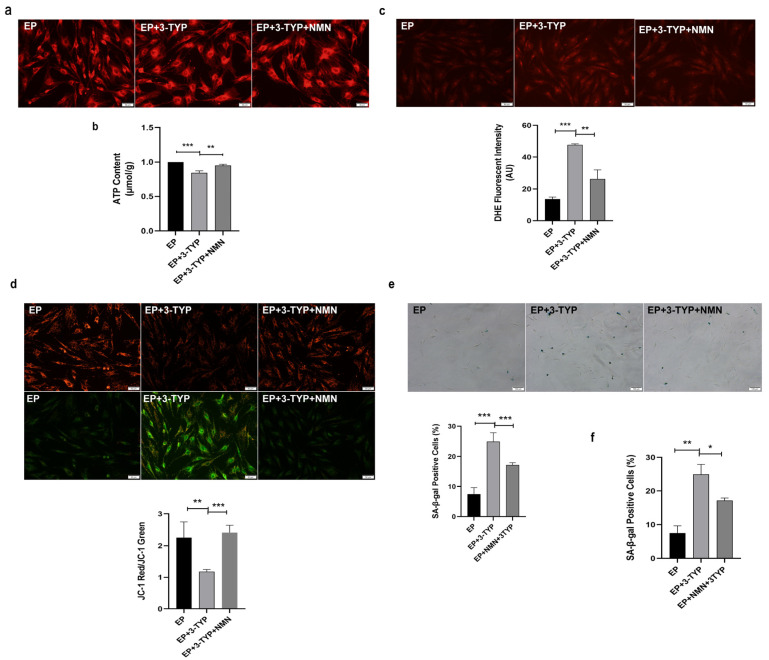
Nicotinamide mononucleotide (NMN) mitigates 3-TYP-induced mitochondrial dysfunction and cellular senescence in early passages mesenchymal stem cells (EP MSCs). (**a**) Mitochondrial morphologic characteristics (scale bar = 50 μm). (**b**) Adenosine triphosphate (ATP) content. (**c**) Intracellular reactive oxygen species (ROS) levels were determined using dihydroethidium (DHE) staining (scale bar = 50 μm). (**d**) JC-1 staining (scale bar = 50 μm) was used to detect mitochondrial membrane potential (MMP). (**e**) Senescence-associated β-galactosidase (SA-β-gal) staining (scale bar = 100 μm) and quantitative analysis of the positive cell rate. (**f**) Examination of P16^INK4a^ mRNA expression by RT-qPCR. Data are expressed as mean ± SD, *n* = 3, * *p* < 0.05, ** *p* < 0.01, *** *p* < 0.001.

**Figure 9 ijms-23-14739-f009:**
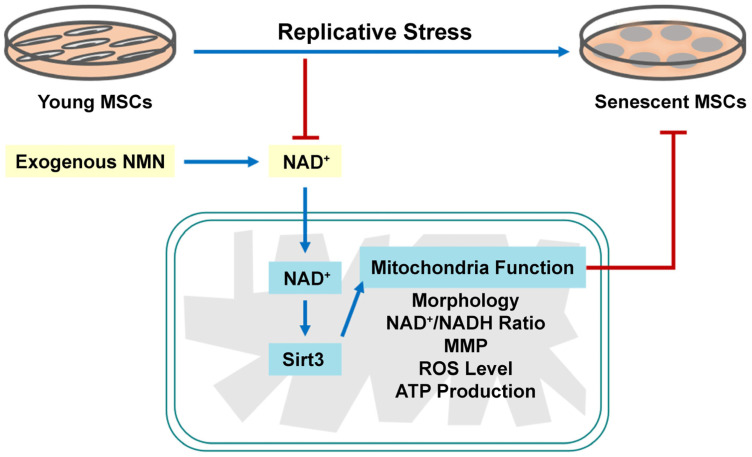
Mesenchymal stem cells (MSCs) exhibited typical senescence-like characteristics as expansion in vitro. Exogenous nicotinamide mononucleotide (NMN) supplementation can ameliorate mitochondrial dysfunction and alleviate MSC senescence through the nicotinamide adenine dinucleotide (NAD^+^)/Sirt3 signaling pathway.

## Data Availability

Not applicable.

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
