# Peer review of "Nicotinamide Mononucleotide Supplementation Improves Mitochondrial Dysfunction and Rescues Cellular Senescence by NAD+/Sirt3 Pathway in Mesenchymal Stem Cells"

_ijms, 2022, doi:10.3390/ijms232314739_

Round 1

Reviewer 1 Report

Dear Authors,

I have carefully reviewed your manuscript. I think it could be accepted after minor revision.

My comments and recommendations are followings:

1. The title "NMN supplementation...  " The title cannot start from abbreviations;

2. List of all abbreviations should be added;

3. The font size in the pictures is unreadable. It is so small. Please increase the the quality of the figures, especially in the graphs.

Best wishes,

Reviewer

Author Response

Response to Reviewer 1 Comments

I have carefully reviewed your manuscript. I think it could be accepted after minor revision.

My comments and recommendations are followings:

  1. The title "NMN supplementation... " The title cannot start from abbreviations;

Response 1: Thank you for your valuable comments. Based on your suggestions, “NMN” has been replaced by “Nicotinamide mononucleotide” replaced in the revised version.

  1. List of all abbreviations should be added;

Response 2: Thank you for your comments. According to your suggestion, list of all abbreviations have been added in the line 609.

  1. The font size in the pictures is unreadable. It is so small. Please increase the the quality of the figures, especially in the graphs

Response 3: We thank the reviewer for pointing out our weakness. As you requested, we have improved the quality of the figures to ensure the font in the pictures is readable.

Reviewer 2 Report

In this manuscript, Wang H et al. showed that late passage (LP) mesenchymal stem cells (MSCs) reduce expression levels of Sirt3 resulting in mitochondrial dysfunction and occurrence of replicative senescence. They also showed that NMN (NAD+ precursor) treatment restored Sirt3 expression and improved mitochondrial function in LP MSCs. Although manuscript is well written, several points need to be verified for publication

Major points

1;  In Figure 2cThey only show RT-qPCR of Sirt3 in EP and LP MSCs. Because Sirt4 and Sirt 5 can compensate for Sirt3 function, show RT-qPCR of Sirt4 and Sirt5 mRNA.

2;  They only did experiments of Sirt3 overexpresson in MSCs. Same experiments as Figure 5 should do in Sirt3 knockdown or CRISPR/Cas9-mediated Sirt3 knockout MSCs.

3; The concentration of 3-TYP they used in Figure 6 - 8 is very high. Because IC50 of 3-TYP for Sirt3 inhibition is 16nM, I recommend to use 3-TYP as 10-50nM.

Minor points

Change “Figure1a” to “Figure 1a” in line 101.

Change “Figure 1H” to “Figure 1h” in line 130.

Change “Figure 3E” to “Figure 3e” in line 180.

Describe the official name of “TMRE” and “TMRM” in line 358.

Author Response

Response to Reviewer 2 Comments

In this manuscript, Wang H et al. showed that late passage (LP) mesenchymal stem cells (MSCs) reduce expression levels of Sirt3 resulting in mitochondrial dysfunction and occurrence of replicative senescence. They also showed that NMN (NAD+ precursor) treatment restored Sirt3 expression and improved mitochondrial function in LP MSCs. Although manuscript is well written, several points need to be verified for publication

Major points

1;  In Figure 2c、They only show RT-qPCR of Sirt3 in EP and LP MSCs. Because Sirt4 and Sirt 5 can compensate for Sirt3 function, show RT-qPCR of Sirt4 and Sirt5 mRNA.

Response 1: Thanks for your constructive comments and suggestion. We agree with your comments that Sirt4 and Sirt5 can compensate for Sirt3 function. SIRT3, SIRT4, and SIRT5 predominantly localize at mitochondria, commonly referred to as mitochondrial sirtuins, and have been proposed to function as a link between aging and metabolism. It is reported that SIRT5 and SIRT3 protect cells from oxidative stress, while SIRT4 exacerbates oxidative stress. Their roles and interactions within mitochondria are quite complicated. So further research will be critical for exploring the compensation function among mitochondrial sirtuins and continuing to approach actual clinical practice. Really thanks for your valuable advice, which provides us with the research interests in the future.

2;  They only did experiments of Sirt3 overexpresson in MSCs. Same experiments as Figure 5 should do in Sirt3 knockdown or CRISPR/Cas9-mediated Sirt3 knockout MSCs.

Response 2: We sincerely appreciate your suggestion. We agree with the reviewer’s opinion. Although Sirt3 overexpresson in MSCs can delay cellular senescence, if the experiments of Sirt3 knockdown or CRISPR/Cas9-mediated Sirt3 knockout MSCs were supplemented, the research data would be more sufficent and enhance the reliability of the conclusions. This will be the focus of our future studies. Thanks again to you.

3; The concentration of 3-TYP they used in Figure 6 - 8 is very high. Because IC50 of 3-TYP for Sirt3 inhibition is 16nM, I recommend to use 3-TYP as 10-50nM.

Response 3: Thank you for your comments. We appreciate the reviewer’s comments. First, 3-TYP, a SIRT3-specific inhibitor, was selected to verify the effects of Sirt3 on mitochondrial function and MSC senescence. Although the IC50 of 3-TYP for Sirt3 inhibition is 16nM according to the instructions, our experimental conditions and the duration of action on MSCs were different. In our study, the effect of 3-TYP on the proliferation activity of MSCs was detected by CCK8 kit and the IC50 of 3-TYP on MSCs for 12h was 180.468μM. Given the short duration of action and the characteristics of young and senescent MSCs, we thus chose 100μm as the concentration of 3-TYP in this study.

Minor points

Change “Figure1a” to “Figure 1a” in line 101.

Change “Figure 1H” to “Figure 1h” in line 130.

Change “Figure 3E” to “Figure 3e” in line 180.

Describe the official name of “TMRE” and “TMRM” in line 358.

Response 4: We thank you for pointing out our weakness. As you requested, we have corrected the above errors and described the official name of “TMRE” and “TMRM” in line 392.

Reviewer 3 Report

In the manuscript “NMN supplementation improves mitochondrial dysfunction and rescues cellular senescence by NAD+/Sirt3 pathway in mesenchymal stem cells” the authors Wand et al. describes how mitochondrial dysfunction and cellular senescence of mesenchymal stem cells is alleviates via supplement with NMN by the NAD+/Sirt3 pathway. The manuscript is nicely written, however there a few point to be clarified:

1. Explain each abbreviation at its first used

2. How you calculated exactly the cell aspect ratio, please add in the methods.

3. Method part lentiviral transduction (line 515): Please describe the used vector system, especially which Sirt3 it is expressed, which promoter is used, how the EGFP is incorporated. In addition, describe how the virus particles were generated and whether you used any selection system. If the virus was bought, please cite the source.

4. Please indicate the ultra-structures mentioned in the result part in the electron microscopy pictures

5. Specificy of DHE should always be tested with by adding ROS scavengers (e.g. PEG-SOD) at least in one experiment as proof of principle. In addition, please indicate a unit for the fluorescence intensity.

6. Explain the abbreviations in the figure legend.

7. Please quantify the MMP

8. Please indicate the sample size for each quantification in the figure legend.

Author Response

Response to Reviewer 3 Comments

In the manuscript “NMN supplementation improves mitochondrial dysfunction and rescues cellular senescence by NAD+/Sirt3 pathway in mesenchymal stem cells” the authors Wang et al. describes how mitochondrial dysfunction and cellular senescence of mesenchymal stem cells is alleviates via supplement with NMN by the NAD+/Sirt3 pathway. The manuscript is nicely written, however there a few point to be clarified:

  1. Explain each abbreviation at its first used

Response 1: We sincerely appreciate your suggestion. Based on your suggestions, we have expained each abbreviation at its first used in the modified manuscript.

  1. How you calculated exactly the cell aspect ratio, please add in the methods.

Response 2: Thank you very much for your comments. We agree with the reviewer’s opinion. Firstly, MSC morphology was observed under a high magnification (´200) microscope, and then multiple fields were selected to collect images. Next, the cell surface area and the longest and the shortest diameters passing through the nucleus of a single MSC were calculated by CellEntry software. Then the ratio of the longest diameter to the shortest diameter was the cell aspect ratio. Finally, the obtained cell surface area and cell aspect ratio were analyzed. The above statements have been added to the “Materials and Methods” section in the revised version.

  1. Method part lentiviral transduction (line 515): Please describe the used vector system, especially which Sirt3 it is expressed, which promoter is used, how the EGFP is incorporated. In addition, describe how the virus particles were generated and whether you used any selection system. If the virus was bought, please cite the source.

Response 3: We appreciate the reviewer’s comments. Commercial lentiviral particles encoding Sirt3 and its control LV-Vector were purchased from GeneChem company. We have cited the virus source in the modified manuscript.

  1. Please indicate the ultra-structures mentioned in the result part in the electron microscopy pictures

Response 4: Thanks to you for your constructive comments and suggestion. We agree with the reviewer’s opinion. Based on your suggestion, we indicated the ultra-structures mentioned in the result part with red frames in the electron microscopy pictures (Figure 1e).

  1. Specificy of DHE should always be tested with by adding ROS scavengers (e.g. PEG-SOD) at least in one experiment as proof of principle. In addition, please indicate a unit for the fluorescence intensity.

Response 5: Thank you for your constructive suggestion and pointing out our weakness. We agree with your opinion. As you said, adding ROS scavengers as proof of principle not only tests the specificity of DHE but also improves the integrity of the experimental results. In future studies, we will use ROS scavengers as proof of principle based on your suggestion. Thanks again for your valuabe advice. Moreover, the fluorescence images were captured using fluorescence microscopy (excitation 300 nm and emission 610 nm) (OLYMPUS, Japan). According to your suggestion, a unit for the fluorescen intensity has been added in the modified figures in the new manuscript.

  1. Explain the abbreviations in the figure legend.

Response 6: Thank you for your comments. Based on your suggestion, we have explained the abbreviations in the figure legend in the modified version.

  1. Please quantify the MMP

Response 7: Thank you for your suggestion. Based on your suggestion, JC-1 red/ JC-1 green (MMP) was quantified by using ImageJ software. The quantitative analyses have been integrated into the modified figures in th new manuscript.

  1. Please indicate the sample size for each quantification in the figure legend.

Response 8: We thank the reviewer for the valuable comment. Three samples for each quantitative analysis were applied in this study, and “n=3” was added in the figure legend. We sincerely hope that you will now consider our revised manuscript worthy of publication.

Reviewer 4 Report

The work presented on the manuscript is very relevant and the data is well structured. No further comments and/or improvements. 

Author Response

Response to Reviewer 4 Comments

The work presented on the manuscript is very relevant and the data is well structured. No further comments and/or improvements.

Response: We are truly thankful for your considerations. The present study systematically elucidated that replicative senescent MSCs exhibit mitochondrial dysfunction, including abnormal mitochondrial morphology and structure, decreased intracellular ATP content, increased ROS level, MMP loss, and reduced OCR. NMN repletion could rejuvenate mitochondrial function and attenuate MSC senescence by activating NAD+/Sirt3 signaling pathway. Thanks again to you.

Round 2

Reviewer 2 Report

I understood their replies, but if they don't do any experiments, describe these topics in the Introduction or Discussion.

Author Response

Response to Reviewer 2 Comments

I understood their replies, but if they don't do any experiments, describe these topics in the Introduction or Discussion.

Response 2: We sincerely appreciate your suggestion. We agree with the reviewer’s opinion. Based on your suggestion, we have added these topics in the Discussion. Thanks again to you.